

# Use of intravenous immunoglobulin in antiphospholipid antibody positive patients with high risk of miscarriage: a systematic review and meta-analysis

Xin Yuan, Wei Zhang, Tong Wang, Peng Jiang, Zong-kui Wang and Chang-qing Li

Institute of Blood Transfusion, Chinese Academy of Medical Science & Peking Union Medical College, Chengdu, Sichuan, China

## ABSTRACT

**Objective:** The purpose of the present study was to evaluate whether intravenous immunoglobulin (IVIG) increases live birth rates and improves neonatal results in patients with antiphospholipid antibodies (aPL) at high-risk for miscarriage.

**Background:** Positivity of aPL in pregnant women is a high-risk factor for miscarriage, and IVIG treatment has emerged as a potential intervention.

**Methods:** The Preferred Reporting Items for Systematic Reviews and Meta-Analyses (PRISMA) guideline was employed to search multiple electronic databases for articles published until August 20, 2023, including PubMed, Web of Science, Embase, Scopus and Medline. The inclusion criteria encompassed studies assessing the efficacy of IVIG in aPL-positive patients with a high risk of miscarriage. Relevant articles were assessed for the quality and data were extracted for analysis. Two independent reviewers performed study selection, data extraction, and quality assessments. The risk of bias was evaluated according to the Cochrane risk of bias tool. All analyses were conducted using Review Manager 5.3.

**Results:** This systematic review included nine randomized controlled trials, with 366 aPL-positive women at high risk of miscarriage. These studies included in this review were randomized controlled trials. The primary outcome measures were successful pregnancy outcomes and live birth rates. The secondary outcomes included obstetric complications, and neonatal outcomes (such as birth weight and live-birth rate). The comparison between the intervention and control groups revealed no significant differences in terms of obstetric complications and neonatal outcomes. The group receiving IVIG treatment had a higher prevalence of preterm deliveries than controls (OR = 2.05, $I^2$ = 46%, 95% CI [0.58–5.24]), but also exhibited a partial improvement in live birth rates (OR = 2.86, $I^2$ = 52%, 95% CI [1.04–7.90]), because it reduced the number of miscarriages (OR = 0.35, $I^2$ = 52%, 95% CI [0.13–0.96]).

**Conclusion:** Based on the available evidence, IVIG intervention appears to be a potentially effective approach for managing of aPL-positive pregnant women with high risk of miscarriage. While IVIG shows significant potential in tripling the chances of having a live-born child, further large-scale randomized controlled trials are necessary, preferably comparing IVIG with hydroxychloroquine or lifestyle and dietary interventions, to refine treatment protocols and ensure the most effective application.

Corresponding authors
Zong-kui Wang,
zongkui.wang@ibt.pumc.edu.cn
Chang-qing Li,
lichangqing268@163.com

## INTRODUCTION

Antiphospholipid antibodies (aPL) are autoantibodies targeting negatively charged phospholipids on platelet and endothelial cell membranes. These antibodies include lupus anticoagulant, anticardiolipin antibodies, and anti-β2 glycoprotein antibodies, which can be detected in individuals with various autoimmune disorders (*Favaloro & Pasalic, 2023*; *Miyakis et al., 2006*). APL-positive hypertensive disorders of pregnancy patients at high-risk for miscarriage include women with triple antibody positivity (lupus anticoagulant, anticardiolipin, and anti-β2 glycoprotein I), a systemic lupus erythematosus diagnosis, previous vascular thrombosis, previous adverse pregnancy outcomes or low complement levels (*Antovic et al., 2018*). Women with aPL exhibit an increased susceptibility to miscarriage, preeclampsia, eclampsia, and stillbirth due to placental insufficiency (*American College of Obstetricians and Gynecologists' Committee on Practice Bulletins—Obstetrics and the Society for Maternal-Fetal Medicine, 2019*; *Xu et al., 2022*). Over 60% of the mothers with positive antiphospholipid antibody who miscarry will have a subsequent miscarriage (*Alijotas-Reig et al., 2019*; *Erton et al., 2022*; *Zhou et al., 2019*). Consequently, the management of aPL-positive individuals at high risk of miscarriage has been a significant challenge for clinicians. To enhance the chances of successful live birth, various treatments have been employed. Currently, the recognized therapeutic agents to improve pregnancy outcomes include aspirin, low molecular weight heparin, hydroxychloroquine, prednisone, and immunoglobulin. Numerous studies have shown that aPL-related pregnancy loss can be prevented by treatment with prednisone combined with low-dose aspirin (LDA) or subcutaneous heparin alone or in combination with LDA (*Kutteh, 1996*). However, the risk of serious pregnancy complications in these patients remains high. Especially, several studies have found combination prednisone and LDA were ineffective in preventing pregnancy loss (*Clark et al., 1999*; *Cowchock et al., 1992*). Furthermore, when aPL-positive patients present with concurrent comorbidities, such as SLE, comprehensive trials and studies become imperative.

Intravenous immunoglobulin (IVIG) is a medication derived from the plasma of thousands of healthy blood donors. It contains a diverse range of antibodies capable of modulating the immune response and is commonly employed in the treatment of autoimmune and inflammatory diseases (*Chaigne & Mouthon, 2017*; *Kazatchkine & Kaveri, 2001*; *Shoenfeld & Katz, 2005*). As a potential therapeutic intervention, IVIG has been suggested for patients with recurrent miscarriage (*Banjar et al., 2023*), with studies demonstrating the utilization of IVIG in the first trimester in patients with antiphospholipid syndrome (APS) to prevent recurrent miscarriages (*Arnout et al., 1994*; *Clark, 1999*; *Kwak et al., 1995*). Most of these studies explored IVIG as an early pregnancy intervention for patients with APS, serving as an alternative to heparin. The IVIG offers the advantage of reduction in the significantly elevated risk of preeclampsia in patients with APS

(*Clark et al., 1999*). In contrast to heparin, IVIG does not increase the risk of bone loss in patients with hypertension or major bleeding. IVIG is believed to exert its therapeutic effects in APS through multiple mechanisms, including neutralizing pathogenic autoantibodies, modulating immune cell activity, and suppressing pro-inflammatory cytokines. Additionally, IVIG may improve endothelial function and regulate the fibrinolytic system. These combined actions help to restore immune system balance and enhance placental function, ultimately improving pregnancy outcomes in women with APS (*Chen & Giles, 2010*; *Hoirisch-Clapauch, 2024*; *Hoxha et al., 2022*). In addition, two small open studies (*Francioni et al., 1994*; *Schroeder et al., 1996*) have suggested that IVIG temporarily reduces clinical and serologic indicators of SLE activity. According to current European Alliance of Associations for Rheumatology (EUCAR) and American College of Rheumatology (ACR)s guidelines, IVIG is recommended in selected cases of obstetric APS, particularly for patients with recurrent pregnancy loss or those unresponsive to conventional therapies, despite the low level of evidence supporting these recommendations (*Andreoli et al., 2017*; *Banjar et al., 2023*). However, the available evidence regarding the efficacy of IVIG treatment for aPL-positive patients with high-risk miscarriage remains limited and, in some cases, contradictory. This systematic review aims to evaluate whether IVIG can prevent obstetric and neonatal complications in aPL-positive patients at high risk of miscarriage.

## METHODS

This systematic review adhered to the PRISMA guidelines, and the present protocol was registered in the PROSPERO database as CRD42023447838.

### Design and search strategy

A comprehensive search was conducted across multiple databases, including PubMed, Web of Science, Embase, Scopus and Medline, to identify relevant studies published between 2000 and 2023. The search terms employed were "pregnancy loss" OR "Abortions, Spontaneous" OR "Miscarriage" AND "antiphospholipid antibodies" OR "aPL" AND "Antibodies, Intravenous" OR "Intravenous Immunoglobulin" OR "IVIG" OR "Immunoglobulins, Intravenous" AND "Randomized Controlled Trial". Studies that evaluated the efficacy of IVIG intervention in aPL-positive patients with high risk of miscarriage were included. In addition to the electronic database search, a manual search of the reference lists of the included articles were performed. Duplicate studies identified from different electronic databases were removed and managed using EndNote software (version X20). The methodology of study selection is illustrated in Fig. 1.

### Eligibility criteria

(1) Studies regarding randomized controlled clinical trials in English were included.

(2) In the trial, aPL-positive patients with a history of three or more consecutive miscarriages were eligible. There were no restrictions on age, race, or course of disease.

(3) The end point data of the literature study was complete.

### Exclusion criteria

(1) Summary, reviews and meta-analyses were excluded.

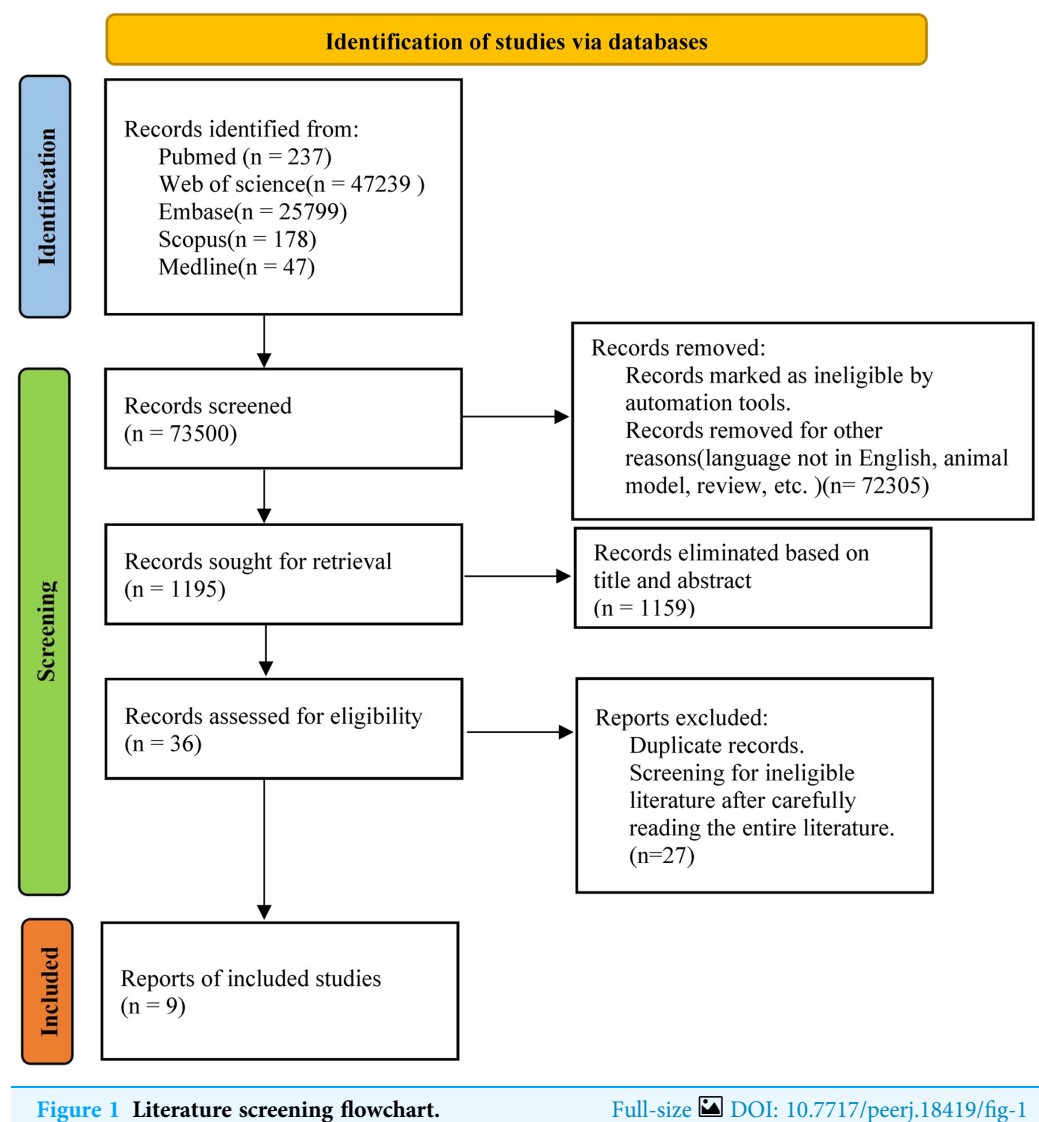

**Figure 1 Literature screening flowchart.**

(2) Studies containing duplicates or insufficient data were excluded.

## Main outcome(s)

Live-birth rate (gestational age (GA) ≥ 37 weeks).

## Secondary outcome(s)

Pregnancy loss (*i.e.*, miscarriages when GA < 20 weeks and stillbirths when GA ≥ 20 weeks), preterm delivery, neonatal outcomes (infants admitted to neonatal intensive care unit, *etc.*), small for gestational age babies, and obstetric complications (gestational diabetes, hypertensive disorders of pregnancy, preeclampsia, *etc*).

## Data extraction

Two authors independently extracted data (XY and WZ). Any discrepancies between them were resolved by discussion or adjudicated by a third author (ZKW). The following data

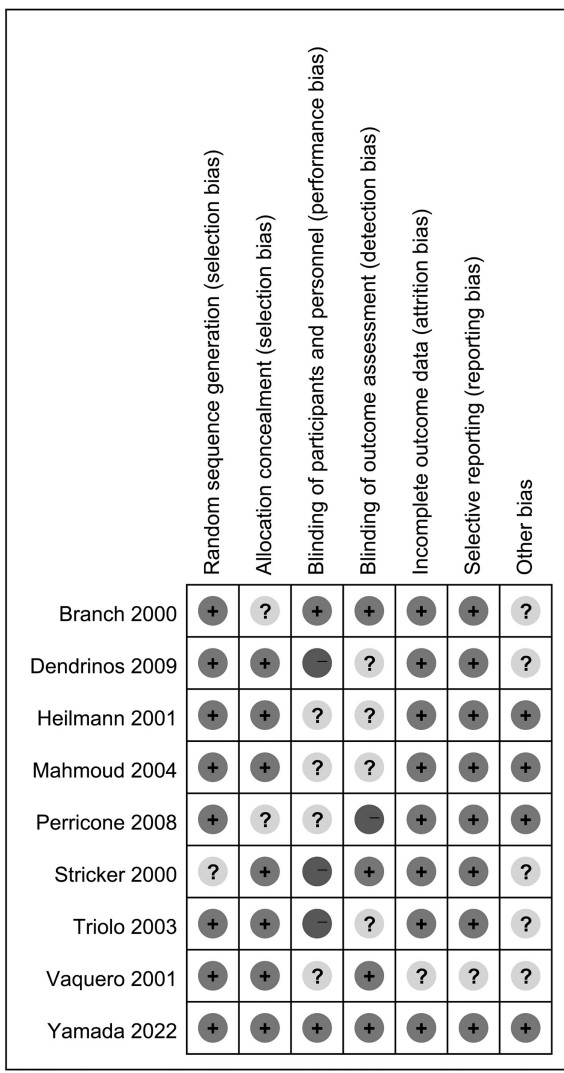

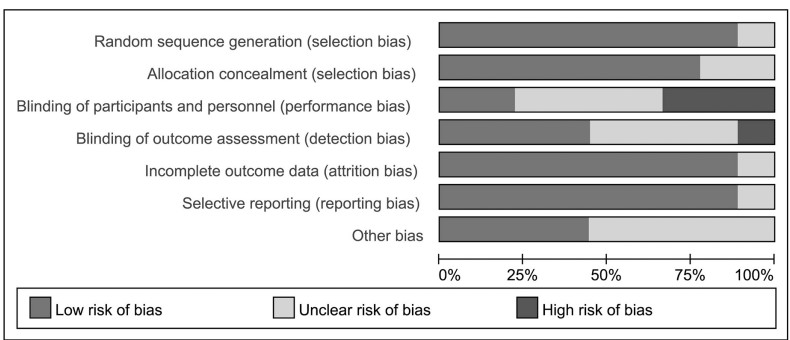

**Figure 2 Assessment for risk of bias in included studies.** Above: The bias assessment conducted for the included studies. Below: The Cochrane Collaboration Risk of Bias Assessment Tool evaluates the risk of bias.

were extracted: (1) data covering author, year of publication, country of origin, trial period, and sample size; (2) participant characteristics including age and intervention specifics such as dosage, frequency, first time of infusion (before pregnancy or gestational week),

**Table 1 Characteristics of included studies.**

| Source | Country of origin | Study design | Trial size | Participant age (years) | Intervention (doses and placebo) | First time of infusion | Number of IVIG infusions | Trial period | Pregnancy outcome after intervention | aPL | Type of adverse pregnancy outcomes before intervention | Risk of bias |
|---|---|---|---|---|---|---|---|---|---|---|---|---|
| Branch et al. (2000) | USA | RCT | 16 | 28.8 ± 3.8 vs. 28.7 ± 4.5 | 10% IVIG 1 g/kg vs. 5% albumin solution two consecutive days each month | As soon as the presence of a live embryo or fetus was confirmed by ultrasonography | 7 | Until 36 weeks' gestation | Seven Live-born infants | +(16) | Recurrent miscarriages, late fetal losses | L, U, L, L, L, L, U |
| Stricker et al. (2000) | USA | RCT | 31 | ≥28 | IVIG 0.2 g/kg every 4 weeks vs. No treatment | After conception occurred | 24 | 26–30 weeks of gestation | 22 had a term pregnancy and two miscarried | +(15) | Recurrent miscarriages | U, L, H, L, L, L, U |
| Mahmoud et al. (2004) | USA | RCT | 15 | Unknown | IVIG 0.5 g/kg intravenously daily vs. Multivitamins for 5 days every month | Once the patients had a positive pregnancy test | 7 | Until about 34 weeks of gestation | Two abortions, none preterm delivery | +(15) | Recurrent miscarriages | L, L, U, U, L, L, L |
| Yamada et al. (2022) | Japan | RCT | 99 | Unknown | 5% IVIG 0.4 g/kg vs. Physiological saline 8 mL/kg administered by intravenous drip infusion for five consecutive days. | Between 4 weeks 0 days and 6 weeks 6 days of gestation after gestational sac was identified | 50 | 5 days after injection | 29 women gave live birth, 19 had miscarriages, one had stillbirth, and one had unknown outcome | +(99) | Recurrent miscarriages, stillbirth | L, L, L, L, L, L, L |
| Perricone et al. (2008) | Italy | RCT | 24 | 34.67 ± 4.27 vs. 34.92 ± 3.53 | IVIG 0.5 g/kg every 3 weeks over a 6-h infusion vs. Prednisolone 0.25–0.5 mg/kg and aspirin (100 mg daily) | As soon as pregnancy was confirmed | 12 | The 33rd week of pregnancy | Pregnancy outcome was successful in all patients treated by means of IVIG Full-term birth (9) Preterm delivery (3) | +(4) | Recurrent miscarriages | L, U, U, H, L, L, L |
| Vaquero et al. (2001) | Italy | RCT | 82 | 31.9 ± 4.7 vs. 30.5 ± 5.1 | IVIG 0.5 g/kg for two consecutive days, once a month vs. Prednisone (15–20 mg till 28th week, 10–15 mg till 32nd week) combined with a daily dose of 100 mg of aspirin. | The 5th week of pregnancy | 53 | The 32nd week of pregnancy | 41 had successful pregnancies and 12 miscarried | +(82) | Recurrent miscarriages | L, L, U, L, U, U, U |

| Source | Country of origin | Study design | Trial size | Participant age (years) | Intervention (doses and placebo) | First time of infusion | Number of IVIG infusions | Trial period | Pregnancy outcome after intervention | aPL | Type of adverse pregnancy outcomes before intervention | Risk of bias |
|---|---|---|---|---|---|---|---|---|---|---|---|---|
| Triolo et al. (2003) | Italy | RCT | 40 | 18–39 | IVIG 400 mg/kg/day given for two consecutive days followed by a single dose each month vs. Low-dose aspirin (75 mg daily) and heparin (self-administered injection; 5,700 IU/day) | As soon as patients had a positive result on a pregnancy test | 21 | 31 weeks' gestation or at the time of miscarriage (IVIG) 34 weeks' gestation or at the time of miscarriage | 12 live births, seven spontaneous abortions one intrauterine death one preterm deliveries one Infants admitted to NICU two Fetal loss after 13 weeks | +(40) | Recurrent miscarriages, late fetal losses | L, L, H, U, L, L, U |
| Dendrinos, Sakkas & Makrakis (2009) | Greece | RCT | 78 | 18–39 | IVIG 400 mg/kg every 28 days vs. 75 mg of low-dose aspirin and 4,500 IU of heparin | As soon as patients had a positive result on a pregnancy test | 38 | 32 weeks of gestation (IVIG, aspirin) 38 weeks of gestation (heparin) | 15 live births, two intrauterine death 1 preterm deliveries 21 First trimester abortion | +(78) | Recurrent miscarriages, late fetal losses | L, L, H, U, L, L, U |
| Heilmann, von Tempelhoff & Kuse (2001) | Germany | RCT | 102 | 26–43 | IVIG 0.3 g/kg for 5 days, followed by 0.3 g/kg for 3 days every 3 to 4 weeks vs. No treatment (All patients were given additional low-molecular weight heparin (3,000 anti-Xa U certoparin/d) and lowdose aspirin (100 mg/d).) | The fifth to sixth week of gestation | 66 | the 28th to 32nd week of gestation. | 16 live births, one fetal growth retardation five preterm deliveries one miscarried | +(17) | Recurrent miscarriages, fetal growth retardation, preterm delivery | L, L, H, U, L, L, U |

number of infusions and duration of treatment; (3) details of the placebo including substance and pregnancy outcomes after intervention, such as live birth, clinical miscarriage, ectopic pregnancy, induced abortion and stillbirth. Standardized forms developed for this specific study were used.

## Risk of bias assessment

Two investigators (XY and WZ) independently assessed the risk of bias based on the following domains as recommended by the Cochrane Handbook (*Higgins et al., 2011*). The third author (ZKW), served as the referee for resolving any disagreements that could not be settled through discussion between the initial two reviewers. The domains included: 1. random sequence generation; 2. allocation concealment; 3. blinding of participants and personnel; 4. blinding of outcome assessment; 5. incomplete outcome data and its handling; 6. selective reporting of the outcomes; 7. any other biases. The results of bias assessment were presented in Fig. 2 indicating low (L), high (H), or unclear (U) risk of bias for each of the seven items in each trial. Any disagreements were resolved by discussion, involving a third author when necessary.

## Strategy for data synthesis

The study design and demographic characteristics of each included study have been summarized in Table 1, which provides an overview of details such as authors, year of publication, country of origin, trial duration, and trial size. All outcome data were analyzed using RevMan 5.3 software.

## Measures of effect

Dichotomous data were expressed as odds ratio (OR) and 95% confidence intervals (CIs), while continuous data were expressed as the mean difference (MD) and 95% CIs. To assess the heterogeneity among the included studies, Cochran's Q test and the Higgins $I^2$ statistic were employed. If $p \geq 0.10$ or $I^2 \leq 50\%$, it indicates that the heterogeneity among the studies is acceptable, fixed effect model was employed for analysis. Conversely, if $p < 0.10$ or $I^2 > 50\%$, suggesting significant heterogeneity, a random effect model was applied for analysis. Publication bias was analyzed for the total effective rate.

## Analysis of subgroups or subsets

Due to data limitation, neither subgroup nor sensitivity analysis were performed. The meta-analysis presented the statistical results for different clinical presentations.

## RESULTS

### Search characteristics and risk of bias assessment

The search yielded a total of 73,500 articles. After filtering the titles and abstracts, 1,195 articles were obtained and assessed for eligibility, and then duplicates were removed. Based on the eligibility criteria, a final selection of nine studies were enrolled (Fig. 1) (*Branch et al., 2000*; *Dendrinos, Sakkas & Makrakis, 2009*; *Heilmann, von Tempelhoff & Kuse, 2001*; *Mahmoud et al., 2004*; *Perricone et al., 2008*; *Stricker et al., 2000*; *Triolo et al., 2003*; *Vaquero et al., 2001*; *Yamada et al., 2022*). These articles were published between 2000 and

2023, with three originating from the USA (*Branch et al., 2000*; *Mahmoud et al., 2004*; *Stricker et al., 2000*), three from Italy (*Perricone et al., 2008*; *Triolo et al., 2003*; *Vaquero et al., 2001*), and one from Japan (*Yamada et al., 2022*), one from Germany (*Heilmann, von Tempelhoff & Kuse, 2001*) and one from Greece (*Dendrinos, Sakkas & Makrakis, 2009*). Among the selected studies, three addressed pregnancy complications such as gestational diabetes, hypertensive disorders of pregnancy, *etc.* (*Branch et al., 2000*; *Heilmann, von Tempelhoff & Kuse, 2001*; *Vaquero et al., 2001*), four mentioned the status of newborns regarding the need for intensive care after birth (*Branch et al., 2000*; *Stricker et al., 2000*; *Triolo et al., 2003*; *Yamada et al., 2022*), and five analyzed the birth weight of the neonates (*Branch et al., 2000*; *Dendrinos, Sakkas & Makrakis, 2009*; *Perricone et al., 2008*; *Triolo et al., 2003*; *Yamada et al., 2022*). Table 1 presents specific details of the included studies. All analyses were conducted using either random effects model or fixed effects model using Review Manager 5.3. No sensitivity analysis was conducted owing to limited data. Table 1 provides a summary of the results of risk of bias assessment.

### Live births and miscarriage rates

Upon consolidating all the included literature in Review Manager 5.3, an initial analysis revealed no discernible difference between the intervention and control groups (result not shown). It is worth noting that three RCTs excluded patients with SLE (*Branch et al., 2000*; *Dendrinos, Sakkas & Makrakis, 2009*; *Triolo et al., 2003*). Upon excluding these three RCTs, a distinct pattern emerged. Specifically, in cases involving aPL-positive high-risk miscarriage patients with SLE or other autoimmune diseases, IVIG treatment demonstrated a notable increase in live birth rate across the six RCTs ($n = 317$): OR = 2.86, $P = 0.07$, $I^2 = 52\%$, $P < 0.05$, 95% CI [1.04–7.90] (Fig. 3A). Furthermore, a statistical analysis of miscarriage rate in the six RCTs ($n = 317$) indicated that IVIG intervention significantly reduced the miscarriage rate of aPL-positive patients at high risk for miscarriage: OR = 0.35, $p = 0.06$, $I^2 = 52\%$, $P < 0.05$, 95% CI [0.13–0.96] (Fig. 3B).

### Preterm delivery

A comprehensive statistical analysis of the preterm delivery rates of all the nine included RCTs ($n = 307$) unveiled that the IVIG intervention group exhibited a higher preterm delivery rate (OR = 2.05, $p = 0.07$, $I^2 = 46\%$, $P < 0.05$, 95% CI [0.58–5.24], Fig. 3C). This suggested a potential association between IVIG intervention and an increased likelihood of preterm birth in patients.

### Obstetric complications and neonatal outcome

Three of the included RCTs ($n = 135$) addressed maternal pregnancy complications including gestational diabetes, hypertensive disorders of pregnancy, *etc.*, and four RCTs ($n = 112$) mentioned infants birth outcomes such as infants admitted to neonatal intensive care unit. It was founded that no significant associations between IVIG intervention group and placebo group in obstetric complications (OR = 1.67, $p = 0.03$, $I^2 = 72\%$, 95% CI [0.20–13.58], Fig. 3D) and neonatal outcomes (OR = 1.42, $p = 0.16$, $I^2 = 45\%$, 95% CI [0.39–5.23], Fig. 3E). Meanwhile, the analysis of five RCTs ($n = 155$) revealed no difference

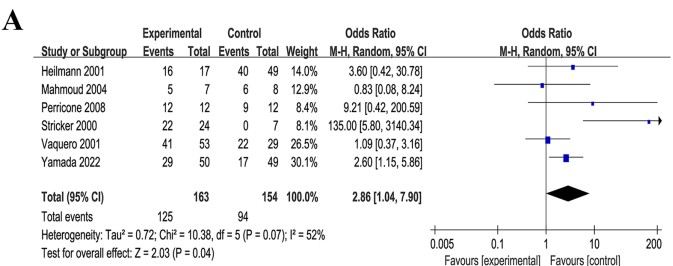

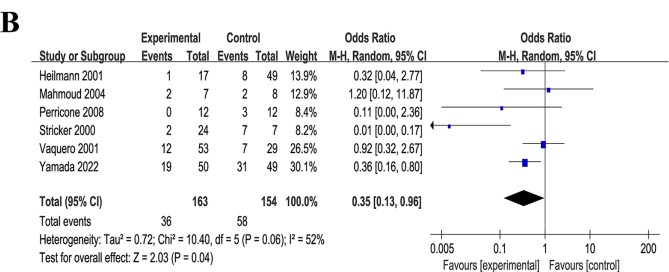

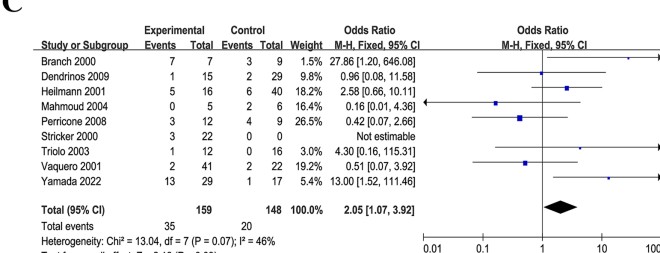

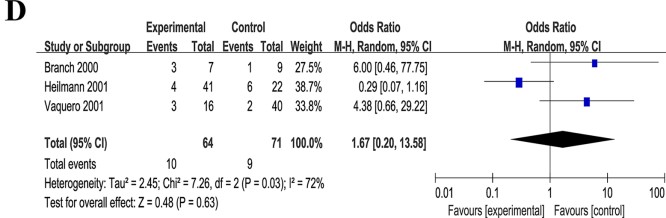

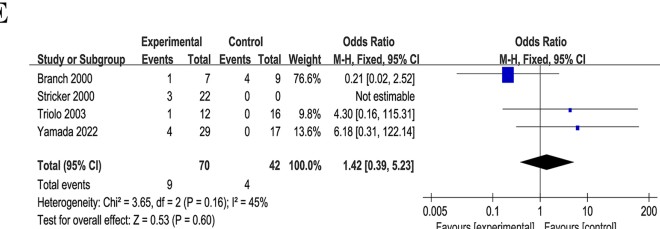

**Figure 3  Forest plots of included studies.** (A) Live birth rates (six comparisons, *n* = 317). (B) Miscarriage rates (six comparisons, *n* = 317). (C) Preterm delivery rates (nine comparisons, *n* = 307). (D) Obstetric complications (three comparisons, *n* = 135). (E) Neonatal outcome (four comparisons, *n* = 112). (CI, confidence interval; MH, Mantel–Haenszel method) (*Branch et al., 2000*; *Dendrinos, Sakkas & Makrakis, 2009*; *Heilmann, von Tempelhoff & Kuse, 2001*; *Mahmoud et al., 2004*; *Perricone et al., 2008*; *Stricker et al., 2000*; *Vaquero et al., 2001*; *Triolo et al., 2003*; *Yamada et al., 2022*).

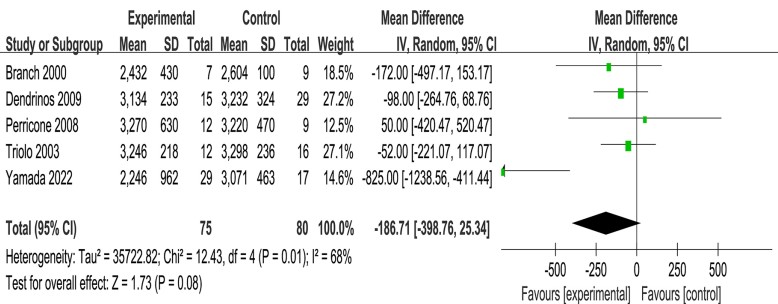

**Figure 4 Forest plots of birth weight (five RCTs, *n* = 155).** (CI, confidence interval; IV, inverse variance method) (*Branch et al., 2000*; *Dendrinos, Sakkas & Makrakis, 2009*; *Perricone et al., 2008*; *Stricker et al., 2000*; *Triolo et al., 2003*; *Yamada et al., 2022*).

in birth weight between the IVIG intervention group and the placebo group (MD = −186.71, $p = 0.01$, $I^2 = 68\%$, 95% CI [−398.76 to 25.34], Fig. 4).

# DISCUSSION

Recurrent miscarriage poses a growing challenge in contemporary society, especially as more and more women delay childbearing into their 30s and 40s. Within this age group, various immune abnormalities affecting successful pregnancy increases. Several studies have established a correlation between pregnancy pathology (such as recurrent miscarriage, preeclampsia and eclampsia) and the presence of anticardiolipin antibodies. Positive aPL typically indicates aPL-associated diseases, such as SLE and APS, which have been proved to be associated with an elevated risk of intrauterine growth restriction, miscarriage, stillbirth and preterm delivery (*Högdén et al., 2019*; *Lam, Brown & Sharma, 2023*). In recent years, advances in treatment during pregnancy have improved outcomes. However, it should be given that fetal and maternal morbidity and mortality remain high. The management of patients who do not respond to conventional therapy in the latter stage of pregnancy poses significant challenges, due to the development of preeclampsia (*Berks et al., 2015*; *Gordon & Kilby, 1998*).

IVIG, successfully employed in a variety of autoimmune disorders, such as Kawasaki disease and idiopathic thrombocytopenic purpura, has been explored as a treatment for aPL-positive patients (*Lam, Brown & Sharma, 2023*; *Li et al., 2023*). *Carreras et al. (1988)* first reported IVIG treatment in patients with lupus anticoagulant positivity and recurrent spontaneous abortion (RSA). Subsequently, several case reports have emerged regarding the treatment of RSA and aPL with IVIG with combination prednisone, or IVIG in conjunction with heparin and aspirin (*Marzusch et al., 1996*; *Stricker et al., 2000*). For high-risk female patients with a history of prior treatment failure, the estimated overall success rate of IVIG intervention was 71% (11 of 17 patients), indicating the potential benefit of IVIG therapy for a specific subset of patients (*Branch et al., 2000*). In pregnancies characterized by severely compromise and growth restriction, IVIG therapy offers a low-risk strategy for reducing autoantibody-mediated disease and improving placental function. *Spinnato et al. (1995)* demonstrated that immunoglobulin treatment during pregnancy resulted in a decrease in anticardiolipin antibody levels in a cohort of women

4000

with APS. Studies on unexplained RSA also suggest a potential role of IVIG in the treatment of recurrent miscarriage. Additionally, with respect to IVIG treatment in RSA patients associated with aPL, the rates of successful live births ranged from 70% to 100%, with a lower incidence of gestational complications compared to traditional protocols (*Branch et al., 2000*; *Clark et al., 1999*; *Harris & Pierangeli, 1998*).

In this review, IVIG was commonly administered at doses ranging from 0.4 to 1 g/kg, initiated early in pregnancy (often after a positive pregnancy test) and continued until 32–36 weeks of gestation. The treatment was frequently combined with other therapies such as low-dose aspirin or heparin. While generally well-tolerated, IVIG's administration varied across studies, highlighting the need for standardized protocols to optimize outcomes in high-risk aPL-positive patients.

In the initial analysis, we included all the screened studies, revealing no significant effects of IVIG on the live birth rate of aPL positive patients (result not shown). Subsequently, upon comprehensive examination of the enrolled RCTs, it was found that three of the RCTs explicitly excluded aPL positive patients with SLE and some other autoimmune disorders. After excluding these three RCTs, the subsequent re-analysis demonstrated varying levels of success in improving live birth rates and reducing miscarriage rates through IVIG intervention in aPL-positive patients with recurrent miscarriage. Furthermore, this effect was more prominent and statistically significant in aPL-positive patients in combination with SLE or other similar autoimmune diseases. Placental angiogenesis is known to be defective in APLA-positive patients, primarily due to reduced fibrinolytic-proteolytic activity. Fibrinolysis is a complex process involving multiple proteins and enzymes, which are crucial for maintaining the balance necessary for proper placental vascular development (*Hoirisch-Clapauch, 2024*). Although the direct evidence linking IVIG's primary mechanisms of action to the fibrinolytic system is limited, it is plausible that IVIG may exert an indirect influence on fibrinolysis through its modulation of immune system inflammatory responses.

IVIG is known to modulate various aspects of the immune system, including the suppression of pro-inflammatory cytokines and the regulation of immune cell activity. This immunomodulation could, in turn, affect the balance of the fibrinolytic system. By reducing inflammation, IVIG might help restore the conditions necessary for effective fibrinolysis and proteolysis, thereby potentially improving placental angiogenesis. While further research is needed to elucidate the precise interactions between IVIG and the fibrinolytic-proteolytic system, this potential indirect effect represents an important area of investigation that could enhance our understanding of IVIG's role in improving pregnancy outcomes in APLA-positive patients (*Andreoli et al., 2017*). The mechanism of action of IVIG in SLE and APS as with other autoimmune diseases appears to be multifactorial, and *Dwyer (1992)* demonstrated this efficacy may be attributed to the presence of anti-unique antibodies in IVIG preparations. These antiidiotypic antibodies manipulate the immune system by neutralizing aPL (unique type) through the formation of a unique anti-unique dimer, thereby enhancing the clearance of aPL. Antiidiotypic antibodies can induce a decrease in aPL production through interaction with B-cell antigen receptors. Additionally, the beneficial effects of IVIG are attributed to the altered structure, function, and dynamics

of the unique-type network that can be restored and returned to normal (*Dwyer, 1992*). Similar mechanisms, involving distinctive type interactions on the surface of T cells (*via* interactions with distinctive type determinants on T cell antigen receptors), may alter T cell function. Likewise, unique type interactions with B cells (*via* the B cell antigen receptor) and as well as the binding of Fc fragment of IgG in IVIG preparations to the Fc receptor (FcgRIIb) may down-regulate B cell proliferation and autoantibody production (*Belina, Spencer & Pisetsky, 2021*). In idiopathic thrombocytopenic purpura, the blockade of Fc receptors by phagocytes prevents the reticuloendothelial system from eliminating platelets and other cells that are coated with autoantibodies. And this phenomenon may also manifest in SLE and APS (*Monnet et al., 2021*; *Nagelkerke & Kuijpers, 2014*). Another potential explanation for the observed effectiveness of IVIG treatment could be its capacity to enhance endometrial receptivity. Dysfunctional immune alterations are involved in procreative failure. The appropriate differentiation and development of the components of the fetal-maternal interface are crucial for successful conception and maintenance of pregnancy. IVIG has been shown potent inhibitory effects on P-selectin–dependent rolling and β2-integrin–dependent adhesion, resulting in decreased leukocyte recruitment and vascular dysfunction in postischemic micro-vessels. Additionally, IVIG regulates pregnancy-related vascular remodeling and trophoblast invasion by modulating decidual NK cells (*Bayry et al., 2023*), potentially promoting embryo implantation. These findings suggest IVIG therapy contributes to a higher rate of successful pregnancies in women with autoimmune disorders.

Regarding other important indicators, such as preterm delivery, neonatal outcomes, our finding indicated that the IVIG-treated group exhibited a higher incidence of preterm labor, of which the underlying mechanisms remain unidentified. Four of the RCTs included in the meta-analysis dealt with neonatal outcomes, and five RCTs assessed birth weight, revealing no significant differences in neonatal outcomes between the IVIG-treated group and the placebo-control group. This suggests that there is no negative impact on the general status of surviving infants and the general vital signs of the infants did not to be affected by prematurity. Furthermore, *Branch et al. (2000)* explicitly indicated that IVIG intervention reduced neonatal admissions to neonatal intensive care unit. Part of the explanation of this phenomenon is that IVIG treatment supplements additional immunoglobulins to the fetus during the early stages when the fetus is unable to produce immunoglobulins independently. Furthermore, despite a high rate of preterm delivery, there are evidence of high live birth rates and low miscarriage rates.

In addition to live birth rates and infant status, evaluating the safety of IVIG treatment in patients with aPL positive autoimmune disorders, who are at a high risk of miscarriage, is crucial in determining the suitability of incorporating IVIG into routine adjuvant therapy. Among the three RCTs that addressed obstetric complications, *Vaquero et al. (2001)* found an increased likelihood of gestational diabetes and hypertensive disorders of pregnancy in women treated with prednisone plus LDA compared to IVIG (14% *vs.* 5%, (3/22 patients) *vs.* (2/41 patients), $P < 0.05$). It is worth noting that IVIG therapy is generally well tolerated, with rare occurrences of side effects. Only one of the included RCTs reported side effects occurred in patients following IVIG therapy, however the side

effects were predominantly mild allergic reactions, such as chest pain, headache, nausea and flushing (*Dendrinos, Sakkas & Makrakis, 2009*). Furthermore, other relevant studies that were not included in this review also indicated that serious side effects did not occur when IVIG used. In fact, most patients experienced no or minimal side-effects, such as flu-like symptoms which could be easily managed with paracetamol (*Cajamarca-Barón et al., 2022*; *Han & Lee, 2018*). More severe side effects such as aseptic meningitis was found to be very rare, and aseptic meningitis is similar to renal failure in that they occur reflecting the formation of immune complexes which usually resolve spontaneously or be managed therapeutically with steroids. The limitations of IVIG therapy include its substantial financial burden and the potential risk of viral transmission. The substantial cost of IVIG therapy may be deemed justified due to its ability to mitigate adverse maternal and fetal complications, which frequently necessitate expensive hospitalization. Nonetheless, IVIG therapy remains one of the safest blood components available for current birth procedures, with no documented cases of viral transmission thus far. In fact, none of the enrolled RCTs reported viral infections in either the mother or the fetus. However, it is important to note that the efficacy of IVIG varies among individuals, and the decision to use IVIG in patients should be made in consultation with healthcare professionals.

Notwithstanding the overall positive results, it is crucial to consider the heterogeneity and limitations among the included studies. Disparities in study design, limited sample sizes and varying dosage may have influenced the obtained outcomes. Additionally, the lack of standardized diagnostic criteria for recurrent miscarriage have further contributed to the heterogeneity. Future studies should aim to address these issues to provide more robust evidence on the efficacy of IVIG therapy. The varying follow-up duration of the included studies in this review also poses challenges in drawing definitive conclusions regarding the long-term efficacy and safety of IVIG treatment in this patient cohort. Therefore, further investigations, particularly large-scale randomized controlled trials with longer follow-up time, is needed to establish the most effective protocol and evaluate the safety and efficacy of IVIG intervention in this specific patient population. However, it is indisputable that IVIG serves as a supplementary or alternative effective therapy for aPL-positive high risk of miscarriage patients combined with SLE or other autoimmune diseases, or for women with side effects or contraindications to heparin and aspirin.

Based on the results of our meta-analysis and the characteristics of the included studies, we propose the following therapeutic algorithm for managing aPL-positive patients at high risk of miscarriage. (1) Initial assessment. Screen for aPL, including lupus anticoagulant, anticardiolipin antibodies, and anti-β2 glycoprotein I. Evaluate the patient's obstetric history and risk factors, such as previous recurrent miscarriages or other pregnancy complications. (2) Standard treatment. Initiate therapy with low-dose aspirin (75–100 mg daily) and low molecular weight heparin as the first-line treatment. (3) Consideration of IVIG. Introduce IVIG therapy (0.4–1 g/kg) in cases where patients have failed standard treatment or exhibit very high-risk factors (*e.g.*, triple antibody positivity, systemic lupus erythematosus, or previous severe obstetric morbidity). IVIG may be administered early in pregnancy and continued until late gestation, depending on the patient's response and risk

profile. (4) Monitoring and adjustment. Regularly monitor pregnancy progress, antibody levels, and response to therapy. Adjust the treatment regimen as necessary based on clinical outcomes and any emerging complications. This therapeutic algorithm positions IVIG as a second-line or adjunctive therapy for aPL-positive patients with a high risk of miscarriage, especially in those who do not respond adequately to standard treatments.

## CONCLUSION

Our meta-analysis suggests that IVIG therapy improves pregnancy outcomes in aPL-positive risk patients with a history of recurrent miscarriage. However, further study is necessary to optimize treatment protocols and reduce heterogeneity among studies. Furthermore, long-term follow-up studies are needed to assess the impact of IVIG therapy on maternal and neonatal outcomes.

### Funding

This work was supported by CAMS Innovation Fund for Medical Sciences (CIFMS, 2021-I2M-1-042), the Science & Technology Department of Sichuan Province (2023NSFSC0714, 2023ZHCG0066 and 2023ZYD0085), and the Scientific Research Project of Sichuan Medical Association (Q22009). The funders had no role in study design, data collection and analysis, decision to publish, or preparation of the manuscript.

### Grant Disclosures

The following grant information was disclosed by the authors:
CAMS Innovation Fund for Medical Sciences: 2021-I2M-1-042.
Science & Technology Department of Sichuan Province: 2023NSFSC0714, 2023ZHCG0066 and 2023ZYD0085.
Sichuan Medical Association: Q22009.

### Competing Interests

The authors declare that they have no competing interests.

### Author Contributions

- Xin Yuan conceived and designed the experiments, performed the experiments, analyzed the data, prepared figures and/or tables, authored or reviewed drafts of the article, and approved the final draft.
- Wei Zhang conceived and designed the experiments, prepared figures and/or tables, and approved the final draft.
- Tong Wang conceived and designed the experiments, analyzed the data, prepared figures and/or tables, and approved the final draft.
- Peng Jiang conceived and designed the experiments, performed the experiments, authored or reviewed drafts of the article, and approved the final draft.
- Zong-kui Wang conceived and designed the experiments, authored or reviewed drafts of the article, and approved the final draft.

- Chang-qing Li conceived and designed the experiments, authored or reviewed drafts of the article, and approved the final draft.

## Data Availability

This is a systematic review/meta-analysis.

## Supplemental Information

Supplemental information for this article can be found online at http://dx.doi.org/10.7717/peerj.18419#supplemental-information.

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
