# Peer review of "Use of intravenous immunoglobulin in antiphospholipid antibody positive patients with high risk of miscarriage: a systematic review and meta-analysis"

_PeerJ, doi:10.7717/peerj.18419_

## Round 0.1 · original submission · Major Revisions

Three expert reviewers have commented extensively. Please carefully address all their concerns - in particular, Reviewer 2 has provided a very detailed list of suggestions and requirements which you must respond adequately to.

Reviewer 1 ·

Basic reporting

In the present study, the authors conducted a systematic review and meta-analysis on the efficacy and safety of intravenous immunoglobulins (IVIg) in women with antiphospholipid antibodies and adverse obstetric outcomes.

The methodology is done according to the PRISMA guidelines, and the protocol has been registered in the PROSPERO database.

Review the use of some references that do not correspond to the text. Examples include #8 and #10. On the other hand, some references are clearly outdated in time such as #46, #47, #48 or #50 and it would be better to update them.

Experimental design

The study is well written, very easy to read, the results are clearly presented, and the figures and the table support the text and facilitate its understanding.

Validity of the findings

1. I understand that this is a limitation of the study and the design of the meta-analysis itself, but I am surprised by the fact that the favourable results of IVIG efficacy are only confirmed in women with another autoimmune disease such as SLE. Should we conclude that the 3 RCTs that exclude women with SLE or another autoimmune disease, include only women with primary APS? And therefore, would IVIG not be effective in this group of women? In this case, how could this difference in efficacy be justified considering that these are immune-mediated diseases in both cases?

2. Another important point to consider and that has not been considered in the data collection is the type of adverse pregnancy outcomes. Within the obstetric morbidity associated with aPL, there is a great deal of heterogeneity, and it is possible that the mechanism that causes recurrent miscarriages is not the same as that that which produces fetal losses. It would be interesting to describe (in Table 1) the type of population in each study in terms of these adverse pregnancy outcomes prior to the intervention.
On the other hand, and although the authors already comment on it in the text, a sub-analysis according to the profile of the previous adverse obstetric outcome or even the aPL profile would be very interesting.

3. In the introduction, I find that there is a lack of reference to the current guidelines of both EULAR and ACR regarding the treatment of obstetric APS and obstetric morbidity associated with the presence of aPL. In fact, both consider the use of IVIG in selected cases, although the level of evidence for these recommendations is low.
In my opinion, the introduction is too long and focuses on types of treatment such as low-dose prednisolone (which are covered by EULAR guidelines only in the subgroup of women with first-trimester abortions).

4. According to the results of the meta-analysis and the characteristics of the included studies, I encourage the authors to propose a therapeutic algorithm and to position the IVIGs in it.

Additional comments

• Review the use of abbreviations, line 73 aPL instead of "antiphospholipid antibodies", line 119 IVIG instead of "intravenous immunoglobulins", line 278 aPL for "antiphospholipid antibodies"

• Línea 86, cambiar “low molecular heparin” por “low molecular weight heparin”

• Line 263, what does "gestational pregnancy" mean?

• Línea 278, añadir “with” tras “in combination”

• Line 282, change "beneficial" to "benefit"

• Line 375, change "it" to "it is"

·

Basic reporting

.....

Experimental design

.....

Validity of the findings

......

Additional comments

Abstract: Please rephrase the objective: The purpose of the present study was to evaluate whether intravenous immunoglobulin (IVIG) increases live birth rates and improves neonatal results in patients with antiphospholipid antibodies (aPL) at high-risk for miscarriage. “

Please inform the readers that APLA-positive patients at high-risk for miscarriage include women with lupus anticoagulant positivity, triple antibody positivity (lupus anticoagulant, anticardiolipin, and anti-β2 glycoprotein I), a systemic lupus erythematosus diagnosis, previous vascular thrombosis, previous adverse pregnancy outcomes or low complement levels (Lupus Science & Medicine 2018;5:e000197.) This information must be provided in the main manuscript.

I do not understand what do you mean with successful pregnancy outcomes. I guess your primary outcome measure was the live-birth rate and the secondary was obstetric complications and neonatal outcomes. Preterm delivery and low birth weight are neonatal outcomes. If this is so, you don’t need to repeat in methods what was already stated in objective. Keep it simple.

Please modify the sentence: “A total of 9 studies were included in this systematic review, encompassing a total of 366 aPL-positive women at high risk of miscarriage. The studies included in this review were randomized controlled trials.”(33 words) to “This systematic review included nine randomized controlled trials, with 366 aPL-positive women at high risk of miscarriage.”(17 words)

Please review the following sentence: “IVIG treatment demonstrated the potential to promote preterm fetal delivery” Perhaps you wanted to say that “The group receiving IVIG treatment had a higher prevalence of preterm deliveries than controls”. Please omit ‘fetal’.

The percentage of preterm deliveries should be calculated from live born children. Let me explain: 100 mothers received IVIG and 100 mothers served as controls. Twenty mothers who received IVIG miscarried. Of the 80 live-born babies, 40 (50%) were premature, 40% of all pregnancies. Sixty controls miscarried. Of the 40 live-born babies, 24 (60%) were premature, 24% of all pregnancies. If we compare the percentage of preterm deliveries taking into consideration all pregnancies, it seems that the intervention increased the risk of preterm deliveries.

Please omit the (g). If the weight difference was not significant, it was not significant in grams, pounds or any other measure.

Please simplify your results in the abstract. (i) If the result was significant, it is < 0.05. You may omit the P value. (ii) If the result was not significant, you don’t need to provide details. (iii) if it increases the live birth rates it’s because it reduced the number of miscarriages.

I suggest: There was no significant difference between intervention and control groups in terms of obstetric complications, such as (preeclampsia? gestational diabetes mellitus? please define), of neonatal outcomes (Again, birth weight and live-birth rate are neonatal outcomes, please define) and birth weight. IVIG treatment doubled the risk of preterm fetal delivery… (please check this information!!) and almost tripled the live birth rate in aPL-positive pregnant women.

Please omit the following sentences from the abstract “The findings of this systematic review suggest that IVIG intervention shows promise in improving successful pregnancy outcomes and live birth rates in aPL-positive patients with high risk of miscarriage. However, it worth noting that IVIG intervention may also contribute to preterm delivery in pregnant women, although no signiûcant disparities were observed in neonatal status.”

Please pay attention to the sentence “Nevertheless, the benefits are somewhat limited, necessitating further studies, especially large-scale randomized controlled trials to establish a standardized protocol for its application”. Do you really think that tripling the chances of having a live-born child is a limited benefit? You may say that large-scale randomized controlled trials are required, preferably comparing IVIG with hydroxycloroquine or with lifestyle and dietary interventions, but you cannot say that the benefits are limited.

Key words: abbreviations such as IVIG and PRISMA cannot be used as key words.

Introduction: Please rephrase: antiphospholipid antibodies can be detected in individuals with or without other autoimmune disorders.

Please omit the following sentence: “The presence of positive antiphospholipid antibodies often indicates antiphospholipid antibody-related (aPL-related) diseases, such as systemic lupus erythematosus (SLE), anticoagulant antibody syndrome (APS), and thrombocytopenic purpura[2].”
APS stands for antiphospholipid syndrome, which can be primary or associated to another autoimmune disease, usually systemic lupus erythematosus.

Please modify the sentence: “Women with aPL exhibit a heightened susceptibility to pregnancy loss, and pregnancies can also be complicated by premature delivery and uteroplacental insufficiency” to “Women with aPL exhibit an increased susceptibility to miscarriage, preeclampsia, eclampsia, and stillbirth due to placental insufficiency”

Please omit the sentence “Several studies have established a relationship between pregnancy pathology (such as recurrent miscarriages, gestational hypertension and preterm delivery) and the presence of anticardiolipin antibodies.” The relationship is not between APLA and gestational hypertension, but between APLA and preeclampsia and eclampsia.

Please modify the sentence: “The likelihood of subsequent pregnancy miscarriage in these individuals has been estimated to exceed 60%”. to “Over 60% of the mothers with positive antiphospholipid antibody who miscarry will have a subsequent miscarriage.”

Please, instead of “pregnancy-induced hypertension syndrome” use “hypertensive disorders of pregnancy”. Avoid abbreviations.

Please, instead of “potential fatal bleeding” use “major bleeding”.

I’m sorry, but obstetric morbidity related to APLA is not due to placental thrombosis. Tissue plasmino­gen activator and plasmin have fibrinolytic activity inside the vessels and proteolytic activity in the extravascular compartment. Proteolysis is essential for angiogenesis. APLA inhibit tissue plas­minogen activator and plasmin, thus affecting placental angiogenesis. Other conditions, such as emotional responses, a high-carbohydrate diet, hyperhomocysteinemia, hypercortisolemia or increased cytokine levels can also impair fibrinolysis-proteolysis, therefore affecting placental angiogenesis. Please refer to: Hoirisch-Clapauch S. The impact of emotional responses on female reproduction: fibrinolysis in the spotlight. In: Seminars in Thrombosis and Hemostasis 2024. Thieme Medical Publishers, Inc.

Please rephrase: “Hence, this systematic review aims to evaluate the feasibility of intravenous immune globulin treatment during pregnancy among aPL-positive patients with high-risk miscarriage and to assess the impact of such treatment on obstetric and neonatal outcomes” to “This systematic review aims to evaluate whether intravenous immune globulin can prevent obstetric and neonatal complications in aPL-positive patients at high risk of miscarriage”

Methods: Please rephrase: “...was registered in the PROSPERO database (registration number
CRD42023447838).” to “...was registered in the PROSPERO database as CRD42023447838.”

Eligibility criteria: When you state that there were no restrictions regarding the number of abortions, do you mean that some of the studies included in your meta-analysis recommended IVIG for women with no previous pregnancy loss? If you state that In the trial, aPL-positive women with history of miscarriages were eligibl, there was a restriction in the number of abortions. The minimum was one. Please exclude there were no restrictions regarding the number of abortions.

Please rephrase “Summary, reviews and meta-analysis were excluded” to “Summary, reviews and meta-analyses were excluded”.

Instead of saying that animal studies and cellular studies had been excluded (do cells miscarry?), please state that “In the trial, aPL-positive women with history of miscarriage were eligible.”

Outcomes: The main outcomes are not usually live birth rates at term. We use take-home babies or live-birth rate. Instead of additional outcomes, please use secondary outcomes. Avoid abbreviations.

Instead of birth weight, please use small for gestational age babies.

Data extraction: When referring to an author within the manuscript, please use the initials.

Result: Use Results instead of Result.

In the sentence “These articles were published between 2000 and 2023, with 3 originating from the USA[23, 24, 29], 3 from Italy[25, 28, 30], and the remaining three from Japan[31], Germany[27] and Greece[26].” you may use 3 or three, not both. It is recommended that we use words from zero to ten, and numerals thereafter. “These articles were published between 2000 and 2023, with three originating from the USA [23, 24, 29], three from Italy [25, 28, 30], and one from Japan [31], one from Germany [27] and one from Greece [26].” Please use words instead of numerals when ten or lower in the whole manuscript. Also, insert a space before each bracket in the whole manuscript.

Instead of “and 5 involved the birth weight(g) of infants”, please use “and five analyzed the birth weight of the neonates”.

Live births and miscarriage rates: Please omit the sentence “The primary objective of the present meta-analysis is to investigate the effectiveness of IVIG intervention in improving the live birth rate in pregnancies of aPL-positive patients at high risk for miscarriage.” as it was already stated in objectives.

Preterm delivery: Again, analyze the percentage of preterm deliveries from the live-born children.

Obstetric complications and neonatal outcome: Please omit the sentence “Complications in the pregnant woman after IVIG administration and the state of the infants were also of concern.”

Discussion: Instead of repeated spontaneous abortions, use recurrent miscarriage.
Please use emerging evidence instead of emerging evidences.

Again, APLA do not correlate with gestational hypertension, but with preeclampsia and eclampsia. Don’t repeat the information, especially when it’s not correct.

I’m sorry, but gestational pregnancy doesn’t exist. You already said that the risk of recurrent miscarriage in these individuals has been estimated to exceed 60%. In discussion you only discuss the results of your study. Please omit everything that doesn’t have a relationship with your findings.

Your discussion begins with “In the initial analysis, we included”

Use APL, aPL, APLA or a-PL, but you cannot use aPL, APL and a-PL.

In discussion, you say that “After excluding these three RCTs, the subsequent re-analysis demonstrated varying levels of success in improving live birth rates and reducing miscarriage rates through IVIG intervention in aPL-positive patients with recurrent miscarriage” Here you say that all studies included women with recurrent miscarriage. Please inform the reader in Methods specifically the minimum number of previous miscarriages to be included in the trials. For most authors, recurrent miscarriage means three consecutive miscarriages.

If you are citing Dwyer et al., include the number of the reference.

Please omit the hyperlink in the whole manuscript. It appears in gray in the pdf.

When you discuss the mechanisms of action of IVIG, you are assuming that the placentas of APLA-positive women display inflammation. This is not what happens. Instead, placental angiogenesis is defective due to reduced fibrinolytic-proteolytic activity. Please refer to Hoirisch-Clapauch S. The impact of emotional responses on female reproduction: fibrinolysis in the spotlight. In: Seminars in Thrombosis and Hemostasis 2024. Thieme Medical Publishers, Inc.

Reviewer 3 ·

Basic reporting

No comments.

Experimental design

No comments.

Validity of the findings

No comments.

Additional comments

Positivity of antiphospholipid antibodies (aPL) in pregnant women is a high-risk factor for miscarriage. In order to evaluate the efficacy of IVIG in aPL-positive patients with a high risk of miscarriage, this systematic review included 9 studies based on The Preferred Reporting Items for Systematic Reviews and Meta-Analyses (PRISMA) guideline. These results indicated that IVIG treatment had the potential to promote preterm fetal delivery, but also exhibited a partial improvement in live birth rates and a reduction in miscarriage rates in aPL-positive pregnant women. The topic is of interest and of valuable, yet several minors need to be revised.
1. There are many formatting errors in the article, please check the article format and ensure the consistency of words, including the contents in line 143, 145, 150, 348,353 and 393. Meanwhile, please check punctuation and spaces, such as in line 230.
2. In this manuscript, most of references are too old, please supplement the latest references to enhance the credibility of this review.
3. In line 146 “There were no restrictions on age, race, course of disease, or number of abortions”, whether these factors affect the therapeutic efficacy of IVIG in aPL-positive patients with a high risk of miscarriage, please provide a detailed explanation.
4. Please provide detailed Figure Legends below each Figure for showing the results.
5. If the article summarizes the administration methods of IVIG in aPL-positive patients with a high risk of miscarriage, it would be more significant.

Annotated reviews are not available for download in order to protect the identity of reviewers who chose to remain anonymous.

---

## Round 0.2 · accepted · Accept

The authors addressed all reviewers and editors comments. Now, the manuscript is ready for publication.

Reviewer 1 ·

Basic reporting

The authors have answered properly my comments

Experimental design

The authors have answered properly my comments

Validity of the findings

The authors have answered properly my comments

Additional comments

The authors have answered properly my comments